

# In-depth characterization of submicron particulate matter inter-annual variations at a street canyon site in Northern Europe

Luis M. F. Barreira[1], Aku Helin[1], Minna Aurela[1], Kimmo Teinilä[1], Milla Friman[1], Leena Kangas[1], Jarkko V. Niemi[2], Harri Portin[2], Anu Kousa[2], Liisa Pirjola[3], Topi Rönkkö[4], Sanna Saarikoski[1], Hilkka Timonen[1]

[1]Atmospheric Composition Research, Finnish Meteorological Institute, Helsinki, FI-00101 Finland
[2]Helsinki Region Environmental Services Authority, Helsinki, FI-00066 HSY, Helsinki, Finland
[3]Department of Automotive and Mechanical Engineering, Metropolia University of Applied Sciences, FI-01600 Vantaa, Finland
[4]Aerosol Physics Laboratory, Physics Unit, Faculty of Engineering and Natural Sciences, Tampere University, Tampere, Finland

*Correspondence to*: Luis M. F. Barreira (luis.barreira@fmi.fi)

**Abstract.** Atmospheric aerosols play an important role in air pollution. Aerosol particle chemical composition is highly variable depending on the season, hour of the day, day of the week, meteorology, and the location of the measurement site. Long measurement periods and high time-resolved data are required in order to achieve statistically relevant amount of data for assessing those variations and evaluate pollution episodes. In this study, we present continuous atmospheric $PM_1$ (particulate matter < 1 µm) concentration and composition measurements at an urban street canyon site located in Helsinki, Finland. The study was performed for four and a half years (2015–2019) and involved highly time-resolved measurements by taking advantage of a suite of online state-of-art instruments such as Aerosol Chemical Speciation Monitor (ACSM), Multi-Angle Absorption Photometer (MAAP), Differential Mobility Particle Sizer (DMPS) and Aethalometer (AE). $PM_1$ consisted mostly of organics, with mean mass concentrations of 2.89 µg m$^{-3}$ (53 % of $PM_1$) followed by inorganic species (1.56 µg m$^{-3}$, 29 %) and equivalent black carbon (eBC, 0.97 µg m$^{-3}$, 18 %). A trend analysis revealed a decrease in BC from fossil fuel ($BC_{FF}$), organics and nitrate over the studied years. Clear seasonal and/or diurnal variations were found for the measured atmospheric $PM_1$ constituents. Particle number and mass size distributions over different seasons revealed the possible influence of secondary organic aerosols (SOA) during summer and the dominance of ultrafine traffic aerosols during winter. The seasonality of measured constituents impacted also on particle´s coating and absorptive properties. The investigation of pollution episodes observed at the site showed that a large fraction of aerosol particle mass was comprised of inorganic species during long-range transport, while during local episodes eBC and organics prevailed together with elevated particle number concentration. Overall, the results enlightened the variability of $PM_1$ concentration and composition in a Nordic traffic site, and its implications on urban air quality. Considering the effects of PM mitigation policies in Northern Europe in the last decades, the results obtained in this study may be considered as illustrative of probable future air quality challenges in countries currently adopting similar environmental regulations.



# 1 Introduction

Increasing concentrations of ambient particulate matter (PM) due to fast industrialization, urbanization and economic growth have become a worldwide concern during the past decades. PM is recognized as one of the most problematic air pollutants due

to its global adverse effects on human health, climate and ecosystems as well as related economic impacts (e.g. IPCC, 2013; Lelieveld et al., 2015). The World Health Organization (WHO) has estimated that, globally, 4.2 million premature deaths were attributable to outdoor ambient air pollution in 2016 (WHO, 2018), from which PM may play an important contribution even at low levels (e.g. Feng et al., 2016). Atmospheric particles can also impact on the Earth's climate (Pöschl, 2005) and they represent one of the largest sources of uncertainty in predicting future climate change (IPCC, 2018), primarily due to the

complex nature of the particles. Furthermore, PM is responsible for detrimental local effects, such as smog production, decrease in visibility, and damages in crops, forests and cultural heritage (e.g. Bonazza et al., 2016; von Schneidemesser et al., 2015; Wang et al., 2009).

Atmospheric PM is a complex mixture of organic and inorganic species, originating from a variety of biogenic and anthropogenic sources, and of black carbon (BC) formed from incomplete combustion of fossil fuels and biomass (e.g.

Goldberg, 1985; Jiang et al., 2019; Zhang et al., 2015). The main constituents of ambient PM are carbonaceous compounds, inorganic ions, mineral dust, trace elements, and water (e.g. Gani et al., 2019; Myhre et al., 2013). The absolute and relative concentrations of these constituents have a large spatial and temporal variation (e.g. Jimenez et al., 2009; Myhre et al., 2013). Atmospheric PM concentration and composition have been increasingly studied in different regions of the world and the mitigation measures for reducing PM concentrations are being under discussion and implementation (e.g.

Chuwah et al., 2016; Davidson et al., 2005).

In general, PM concentrations in Nordic European countries are low. Earlier studies and monitoring performed in Nordic metropolitan areas revealed mean $PM_1$ and $PM_{2.5}$ concentrations of about 5–10 µg m$^{-3}$ (Aurela et al., 2015; EEA, 2019; Teinilä et al., 2019; Vogt et al., 2013). The main local anthropogenic fine particle sources in Nordic cities are residential wood burning, traffic-related emissions, energy production and industrial processes (Korhonen et al., 2019), but pollution episodes originating

from long-range transport can also influence significantly the measured atmospheric aerosols (e.g. Molnár et al., 2017; Niemi et al., 2009). However, previous studies have been lacking either long-term measurement periods or comprehensive chemical characterization of $PM_1$. A few long-lasting campaigns have been performed in Northern Europe, but not in urban areas (e.g. Heikkinen et al., 2020) or only involving measurements of specific elements such as BC (e.g. Luoma et al., 2020).

In this study, we present a continuous in-depth chemical characterization and size distribution

measurements of atmospheric $PM_1$ at a busy street canyon site located in Helsinki, Finland. This study was performed during a period of four and a half years, by employing a variety of instruments for characterizing $PM_1$; an Aerosol Chemical Speciation Monitor (ACSM) for the characterization of non-refractory $PM_1$ (NR-$PM_1$) constituents, a Multi-Angle Absorption Photometer (MAAP) for the measurement of equivalent black carbon (eBC), and an Aethalometer for the source apportionment of eBC. Results are compared with other data concurrently measured at the sampling site, including particle



number and mass concentrations, particle size distributions, atmospheric oxidants ($O_3$, $NO_x$) and meteorological variables. This study focuses on the diurnal, seasonal and inter-annual variation of $PM_1$ concentration, composition and size distributions in a Nordic traffic environment. Results give insights to how the mitigation actions targeted to traffic emissions are affecting the PM constituents in Northern Europe.

## 2 Experimental section

## 2.1 Measurement site

Measurements were carried out in Helsinki, Finland, at an urban Supersite measurement station (street address Mäkelänkatu 50) located in a street canyon and operated by the Helsinki Region Environmental Services Authority (HSY). The measurement site is situated at the kerbside of one of the main streets in Helsinki and represents a typical northern European traffic site. The street consists of six lanes, two rows of trees, two tram lines and two pavements, resulting in a total

width of 42 m (Hietikko et al., 2018). During the measurement campaign, the average number of vehicles driving along the street during weekdays was 28 000 per day and the share of heavy-duty vehicles was 10–12 % (statistics from the City of Helsinki). Apart from maintenance breaks, measurements spanned from June 2015 to December 2019. Figure S1 shows the measurement periods of the different instruments employed in this study.

## 2.2 Measurements of atmospheric aerosol particles

The instruments were installed inside a measurement container. The outdoor air was drawn from the roof of the measurement station (4 m above the ground) through a 50 mm stainless steel tube with an air blower to provide large flow (> 200 L min⁻¹) and reduce particle losses. From that large flow, measurement instruments were taking smaller sample flows as described in the following sections.

### 2.2.1 Aerosol chemical speciation monitor

A quadrupole-aerosol chemical speciation monitor (Q-ACSM, Ng et al. (2011), Aerodyne Research Inc., Billerica, USA) was employed for the characterization of non-refractory $PM_1$ (total organics, sulphate, nitrate, ammonium, and chloride). The ACSM system sampled ambient air with a by-pass pump (3 L min⁻¹). A cyclone was used to remove particles larger than $PM_{2.5}$. A Nafion drier was installed prior to the instrument inlet. The relative humidity (RH) of the sample flow was mainly below 30–40 %. A subsample was then taken by the ACSM at ~ 0.08 L min⁻¹ through a 100 µm diameter critical orifice located at

the inlet of an aerodynamic lens. The aerodynamic lens is used to focus submicron particles into a high vacuum system. The typical transmission range (50 %) of the lens is from 75 to 650 nm (Liu et al., 2007). After the lens, non-refractory particles are flash vaporized at 600 ℃ and ionized by electron ionization (70 eV). The resulting ions are then separated and detected, according to their mass to-charge (m/z) ratios, by a Prisma quadrupole mass spectrometer (Pfeiffer Vacuum, model





QMS220). The detected ions were classified into inorganic and organic fragments using a fragmentation table presented by

Allan et al. (2004).

During data acquisition, filtered and particle-laden air were measured interchangeably by averaging 28 scans, resulting in a circa 30 min time resolution. The filtered signal (gases) are subtracted from the particle-laden signal (particles + gases) yielding only particles signal. The effective nitrate response factor ($RF_{NO3}$) and relative ionization efficiency of sulphate ($RIE_{SO4}$) were determined by performing an instrumental calibration to convert analyte signals into nitrate equivalent mass concentrations

using dried and size-selective (300 nm of mobility diameter) ammonium nitrate and diammonium sulphate particles. An effusive source of naphthalene, located in the detection region, is used as a reference for m/z and ion transmission calibrations. Post-processing of data was performed using ACSM local v. 1.6.1.0 within the Igor Pro v. 6.37 (Wavemetrics, OR, USA). The collection efficiency was calculated according to Middlebrook et al. (2012), by determining a chemical time-dependent collection efficiency to correct particle losses due to bouncing off the vaporizer before flash vaporization (Fig. S2). As an

exception from Middlebrook et al. (2012), a collection efficiency of 0.45 was used for samples when ammonium was below the detection limit. The limit of detection (LOD) for ammonium was determined by adding a HEPA filter to the sampling line, averaging the measured mass concentrations of ammonium using a collection efficiency of 0.5, and multiplying the resulting standard deviation by three. The obtained LOD value for ammonium was 0.232 µg m$^{-3}$.

### 2.2.2 Multi-angle absorption photometer

A multi-angle absorption photometer (MAAP, model 5012, Thermo Fisher Scientific) was used to measure eBC mass concentrations in real-time. MAAP has been described in detail by Petzold and Schönlinner (2004). Briefly, MAAP measures the change in optical transmission as particles are collected on a filter, in combination with the simultaneous measurement of scattered light with multiple detectors to reduce instrumental artefacts and to provide a more accurate measurement of eBC. A mass absorption cross-section value of 6.6 m$^2$ g$^{-1}$ was used to convert the measured light

absorption coefficient value (at wavelength 670 nm) to eBC mass concentration. The flow to MAAP was 11 L min$^{-1}$ and a PM$_1$ inlet was used. The measurements time resolution was set to 1 min. The detection limit of the MAAP, as reported by the manufacturer, is 0.05 µg m$^{-3}$ and the measurement range is 0–180 µg m$^{-3}$ when ten minutes averaging time is used. The measured eBC from MAAP was used to calculate the coating factor (CF) according to Drinovec et al. (2017), which defined CF as the mass of potential material available for the coating of BC particles, expressed as the sum of sulfate, ammonium,

nitrate, and organic mass, divided by the eBC mass.

### 2.2.3 Aethalometer

For the source apportionment of eBC, a dual-spot aethalometer (AE33, Magee Scientific) was used (Drinovec et al., 2015; Hansen et al., 1983). The AE33 allows for the real-time measurement of aerosol particle light absorption and corresponding eBC mass concentrations at seven different wavelengths (370–950 nm), and it compensates for filter loading artifacts and tape

advancement errors when measuring eBC concentrations (Drinovec et al., 2015). The sampling flow rate was set to 5 L min$^{-1}$,





the inlet cut-off size was 1 μm (sharp cut cyclone, BGI model SCC1.197), and the measurement time resolution was set to 1 min. The filter tapes (Magee Scientific) consisted of TFE-coated glass fiber filters and during the long-term measurements different types of tapes were used. From October 2015 to December 2017 a M8020 filter tape was employed (apart from a period from September 2016 to October 2016 when the filter tape was a M8050), while from January 2018 to December 2019

the filter tape was a M8060. As the manufacturers' research report has shown (Magee Scientific, 2017), the M8050 tape could not have been used for eBC source apportionment and thus the data from the two months when this tape was used (autumn 2016) was rejected.

To estimate the contribution from wood burning and fossil fuel to eBC, the source apportionment method referred as the Aethalometer model was employed (Sandradewi et al., 2008). Briefly, the light absorption coefficients measured at

wavelengths of 470 nm and 950 nm were used to calculate the biomass burning percentage (BB), which was then used to determine the $BC_{FF}$ and $BC_{WB}$ concentrations. The fossil fuel absorption Ångström exponent ($\alpha_{FF}$) used in this study was 1.1, while the corresponding exponent for wood burning ($\alpha_{WB}$) was 1.6. These α values have been previously optimized for the sampling site of our study (Helin et al., 2018). For the calculation of $BC_{FF}$ and $BC_{WB}$ mass concentrations, the eBC measured by MAAP was used. In addition, the filter-loading effect compensation parameter ($k$), which in AE33 is used to correct the

light absorption coefficient in real-time and is given directly as data output values, was used for interpreting results.

The manufacturers' research report (Magee Scientific, 2017) has specified that the BB percentage results are 10 % larger with the filter tape M8060 than with the filter tape M8020. In this study, it was estimated that the difference in BB percentage between the filter tape types was approximately 12 %, thus the results after 2018 were post-corrected by applying this correction factor to account for the artificial tape-induced change in the BB percentage (Fig. S3).

**2.2.4 Differential mobility particle sizer**

Particle number size distributions for the sub-micrometer particles were determined by using a differential mobility particle sizer (DMPS; Knutson and Whitby (1975)). DMPS has a differential mobility analyzer (DMA, Vienna-type), for the selection of particle sizes, and a condensation particle counter (CPC, A20 Airmodus) to obtain particle number concentrations for each size bin. The scanned size range in this study was from 6 to 801 nm (mobility diameter, $D_m$). In order to match the sizes of

particles measured by the ACSM and MAAP with the particle sizes measured by DMPS for comparison purposes, the $D_m$ was converted into vacuum aerodynamic diameter ($D_{va}$) that corresponds to the diameter of particles measured by ACSM. For that purpose, the particle density was determined by using the chemical composition data from the ACSM and MAAP and calculating a chemical time-dependent gravimetric density based on the following Eq. (1) from Salcedo et al. (2006):

$$density = \frac{[NO_3 + SO_4 + NH_4 + Org + eBC_{MAAP}]}{\left[\frac{[NO_3] + [SO_4] + [NH_4]}{1.75} + \frac{[Org]}{1.2} + \frac{[eBC_{MAAP}]}{1.77}\right]}. \qquad (1)$$






The calculated mean density was 1.42 g cm$^{-3}$. This mean density value was then employed to convert $D_m$ into $D_{va}$. A time-series of density can be found in the supplementary material (Fig. S2). The density obtained for each data point was subsequently used to convert particle number concentration (PNC) into particle mass concentration (PMC), by selecting $D_m$ between 6 to 549 nm from DMPS measurements (next size bin would be between 549 to 663 nm with the maximum size overcoming the ACSM transmission of up to 650 nm particle size as referred in Sect. 2.2.1).

### 2.2.5 Auxiliary measurements

Basic air quality parameters were measured at the Supersite with the following instruments; nitrogen oxides ($NO_x$) with Horiba APNA-370, ozone ($O_3$) with Horiba APOA-370, and fine particles ($PM_{2.5}$) with TEOM 1405. Temperature (T), RH and precipitation were also measured at the Supersite, while wind speed and wind direction measurements were carried at a meteorological station above roof level (53 meters above the land surface) located approximately 500 meters to north-west from the measurement site. The mixing height was calculated using the model (MPP-FMI) presented by Karppinen et al. (2000). The wind speed and mixing heights were used to evaluate the role of atmospheric dilution in pollutant concentrations by determining the ventilation coefficient (VC), which has been previously used in other studies to characterize dilution of pollutants (e.g. Gani et al., 2019; Sujatha et al., 2016).

### 2.3 Data handling

Most air quality and meteorological parameters were monitored with a 1 min time resolution throughout the corresponding sampling periods (Fig. S1). However, hourly mean or median values were used for data analysis. All data shown is displayed in local time (UTC+2/UTC+3). As required in the EU air quality regulations, all aerosol particle components were calculated to prevailing T and p conditions, while gases are presented in STP conditions (+ 20 °C).

The full time series of individual air quality parameters was employed for most of the data handling. However, since the measurement periods of used instruments differed slightly (Fig. S1), the data captured has been harmonized in some analysis to account for the comparability of results. For example, to calculate the relative contribution of aerosol particle constituents to $PM_{2.5}$, only the concomitant measurement points between TEOM, ACSM, MAAP and Aethalometer were used.

Since the ACSM response was varying over time (Fig. S4a), an additional correction of ACSM was required. For that purpose, monthly mean values of NR-$PM_1$ constituents measured by ACSM and eBC measured by MAAP were used to calculate the mean density per month, which was then employed to determine the monthly mean mass concentration of particles measured by DMPS. The obtained mass concentration was subtracted by eBC and divided by NR-$PM_1$ to obtain monthly correction factors that were employed to ACSM data. A similar correction has been applied also in Heikkinen et al. (2020). The corrected NR-$PM_1$ values were subsequently used to recalculate the particle density and determine the PMC from DMPS measurements as referred in Sect. 2.2.4.

In order to perform a trend analysis for the individual aerosol particle components, the Theil-Sen approach was used (Sen, 1968; Theil, 1950). The Theil-Sen slope estimator was applied by using the TheilSen function of the openair R package in R





software (Carslaw, 2019; Carslaw and Ropkins, 2012; R Core Team, 2020). Monthly mean values were derived by using a data coverage criterion of ≥30 % and the time series were deseasonalized by using the stl (seasonal trend decomposition using
loess) function (Carslaw, 2019). Since the stl function does not allow for missing data, the missing months were linearly interpolated (Carslaw, 2019). Autocorrelation was taken into account in the time series (Carslaw, 2019). The trend analysis output includes the resulting Theil-Sen slope estimate value with the 95 % confidence interval and the corresponding p-value indicating the significance level. Similar approach for trend analysis has been previously used in several studies (e.g. Font and Fuller, 2016; Grange et al., 2020; Henschel et al., 2015; Masiol et al., 2017; Masiol and Harrison, 2015; Munir et al., 2013).
The potential source areas of particles during long-range transport episodes were investigated using back-trajectories produced by the Hybrid Single Particle Lagrangian Integrated Trajectory (HYSPLIT) model (Rolph et al., 2017; Stein et al., 2015).

## 3 Results

Aerosol particle chemical composition and concentration are subject to significant variations over the seasons and years because of e.g. changes in anthropogenic and biogenic activities, meteorology, and political measures to improve air quality.
In this section, we discuss airborne particles chemical composition, concentration, and other critical parameters influencing air quality. A general description of meteorology at the Supersite station during the measurement period (2015–2019) is presented in Sect. 2 of supplementary material.

### 3.1 Trends in chemical composition and concentration of submicron aerosol particles

The mean and median concentrations of $PM_{2.5}$ measured by TEOM and $PM_1$ constituents measured by ACSM and MAAP are
represented in Table 1. The eBC from fossil fuel combustion ($BC_{FF}$) and from wood burning ($BC_{WB}$) measured by aethalometer are also included. For an overview, the monthly median concentrations of organics, sulfate, nitrate, ammonium, eBC, $PM_{2.5}$, particle number concentration (PNC), particle density, and of the most relevant meteorological parameters (T, RH, Wd (wind speed, monthly mean), Ws (wind direction, monthly mean) and gases ($NO_x$ and $O_3$) are shown in Fig. 1.
The measured $PM_{2.5}$ mean concentrations were 21–25 % lower in our study than the concentrations measured between 1999–
2001 and 2013–2015 at urban traffic sites in Helsinki, where the three- and two-year´s $PM_{2.5}$ mean concentrations were 9.6 µg m$^{-3}$ (Laakso et al., 2003) and 9.1 µg m$^{-3}$ (Teinilä et al., 2019) respectively. Similarly, $PM_1$ concentrations were about 25 % lower in our results when compared to the measurements conducted in 2013–2015 (Teinilä et al., 2019). In our study, a decrease of 0.46 µg m$^{-3}$ per year (p <0.05) was observed for $PM_{2.5}$ from 2015–2019 (Fig. 2a). This decrease has been also observed by Luoma et al. (2020) at the same sampling site from 2015–2018. The reduction in $PM_{2.5}$ has been observed in Helsinki during
the last decade, both at traffic and background sites (Laakso et al., 2003; Lorelei de Jesus et al., 2020; Luoma et al., 2020). This $PM_{2.5}$ reduction has been generally observed in European Union (EU) due to decreasing pollutant emissions, which also impacts in the measured concentrations caused by long-range transport (EEA, 2019). Furthermore, the local $PM_{10}$ from street



dust have decreased in Helsinki by employing efficient cleaning methods and a dust binding agent ($CaCl_2$), which also reduces $PM_{2.5}$ (Stojiljkovic et al., 2019).

The sum of NR-$PM_1$ concentrations measured by ACSM and eBC concentrations measured by MAAP was compared to the mass concentration calculated from the DMPS data (Fig. S4b). A correlation of $R^2=0.84$ and a slope of $0.855 \pm 0.002$ were obtained, showing a reasonable agreement between the measurement techniques considering the nature of these long-term measurements that included periods with both high and low mass loadings. The observed differences from the unity may be attributed to uncertainties in the estimation of density for organics and slightly differences in particle size ranges measured by

the ACSM, MAAP and DMPS. A default density value of 1.2 g cm$^{-3}$ has been used for organics, although the density can vary between 0.77–1.90 g cm$^{-3}$ (Turpin and Lim, 2001). In addition, uncertainties associated with the calibration procedure and changes in the transmission efficiency over the size range of measured particles have been previously reported for ACSM analysis (Bressi et al., 2016; Liu et al., 2007) and cannot be excluded in this study. Since most of air quality regulation is based on $PM_{2.5}$ (Council Directive 2008/50/EC, 2008), $PM_1$ from ACSM and MAAP was also compared with $PM_{2.5}$ measured by

TEOM. As shown in Fig. S5, the slope of $PM_1$ vs $PM_{2.5}$ was $0.613 \pm 0.003$, indicating that slightly more than half of $PM_{2.5}$ consisted of $PM_1$ particles.

Organics dominated the $PM_1$ mass concentrations over the measurement period. The trend analysis revealed a statistically significant decrease in organics of 0.26 µg m$^{-3}$ per year (p <0.05, Fig. 2b). Of the inorganic constituents, sulfate was dominating (13 % of $PM_1$), followed by nitrate (10 % of $PM_1$) and ammonium (6 % of $PM_1$). Nitrate decreased at a rate of 0.048 µg m$^{-3}$

per year (p<0.10, Fig. 2c), which might be a consequence of decreasing $NO_x$ emissions at the measurement site (e.g. Luoma et al. (2020) and Fig. 2g). A decrease in ammonium nitrate during the past years as a result of both increasing temperature and reduced emissions has been also reported in another study (Megaritis et al., 2014). Any statistically significant changes in concentration over the years was not observed for the remaining inorganic constituents measured in this study (Figs. 2d and 2e).

Statistically significant decreasing trends were observed for eBC and $NO_x$, with reduction rates of 0.14 µg m$^{-3}$ per year and 11.01 µg m$^{-3}$ per year, respectively (p <0.001, Figs. 2f and 2g). A similar decrease in eBC concentrations of 0.09 µg m$^{-3}$ per year has been found in another study from 2015–2018 (Luoma et al., 2020). In the same study, the decrease in eBC has also been followed by a 11.00 µg m$^{-3}$ yearly decrease in $NO_x$, suggesting the dominant implication of vehicle fleet renewal on the observed eBC reduction. In our study, we also performed a trend analysis for the source apportionment of eBC. The trend

analysis of $BC_{FF}$ and $BC_{WB}$ revealed that traffic was the main driving force behind the decrease in eBC, with decreasing rates of 0.09 µg m$^{-3}$ per year (p <0.001, Fig. 2h). No significant decreasing trend was found for $BC_{WB}$ (Fig. 2i).

It should be noted that, although all studied time series spanned over about four and a half years, the duration of these measurements might still be slightly short for long-term trend analysis. The uncertainties associated to MAAP and ACSM measurements could have also influenced the trend analysis.



## 3.2 Seasonal variation of aerosol particle chemical constituents

The fractional contribution of $PM_1$ constituents is represented in Fig. 3. The monthly median contribution of organics varied between 36 % to 73 %. The dominance of organics was particularly evident during summer, with a relative contribution to $PM_1$ of 64 %. The predominance of organics in $PM_1$ has been reported in many other studies performed at urban sites (e.g. Katsanos et al., 2019; Ripoll et al., 2015).

The seasonal median contribution of eBC to $PM_1$ was relatively constant, varying between 16 % and 18 %. However, the eBC contribution in our study was relatively large when compared to previous studies performed in Helsinki, where a 13 % contribution in winter 2010–2011(1.12.10–07.01.11; Aurela et al., 2015) and a 11 % contribution between 2013–2015 were observed (May, 2013–April, 2015; Teinilä et al., 2019). The relative contribution of eBC was also high when compared to some of the studies performed at urban sites around the word (Fig. S6). This is partially explained by the fact that the site of our study was located beside a busy street. Furthermore, the proportion of heavy-duty vehicles, which are mostly powered by diesel engines, was about 10–12 % at the sampling site. Therefore, these vehicles are likely contributing significantly to the measured eBC concentrations. These evidences depict the importance of local traffic emissions to the currently measured $PM_1$. The monthly variations of the $PM_1$ constituents measured in this study are shown in Figs. 4 and 5. Organics exhibited the largest seasonal variations among all NR-$PM_1$ components (Fig. 4a). The median concentration of organics increased during spring and summer, reaching up to 3.18 µg m$^{-3}$ in July. This increase in organics is likely linked to the enhancement of local SOA formation from biogenic and/or anthropogenic precursors, as a result of photochemical processes in the atmosphere that are predominant during the warmest periods of the year (e.g. Heikkinen et al., 2020). Furthermore, biogenic SOA can also be regionally transported into the sampling site, which will influence the measured concentration of organics. For nitrate and ammonium, a slight decrease in concentrations was observed during summer (Figs. 4b and 4c). Nitrate can be formed by traffic activities through the oxidation of $NO_x$ (e.g. Zhang et al., 2015) but the high volatility of ammonium nitrate causes its dissociation at high temperatures, which can partially explain the decrease of these $PM_1$ constituents during the warmest months of the year together with changes in VC (Fig. S7). On the other hand, lower temperatures and high RH shift the equilibrium towards the particle phase by favoring the thermodynamic gas to particle partitioning of ammonium nitrate (e.g. Zhang et al., 2015). Sulfate had slightly less seasonal variability than organics or nitrate (Fig. 4d), even though higher concentrations were observed during winter.

Interestingly, all inorganics had large variations during winter and early spring. A possible explanation is that, during cold months, there is an intensification of anthropogenic pollution sources of some aerosol particle constituents (e.g. sulfate from coal- and oil-fired energy production and heating sources). This intensification can promote an increase in inorganic particulate composition at both local and distal regions through long-range transport, with the increase being exacerbated by unfavorable meteorological conditions favoring tropospheric accumulation that occur more frequently during winter (e.g. stagnant conditions and shallower mixing layer height) (e.g. Pey et al., 2010). In general, emission mitigation actions have decreased industrial and traffic emissions in Northern Europe during the last decades (e.g. Anttila and Tuovinen, 2010; EEA, 2019;





Hienola et al., 2016), which have resulted in a decrease in atmospheric $SO_2$ and $NO_x$ concentrations. Even though these mitigations must hold in all EU, relatively high inorganic concentrations have been recently measured in other European countries such as in Greece (e.g. Athens, Katsanos et al. (2019)) Spain (e.g. Montsec, Ripoll et al. (2015); Montseny, Minguillón et al. (2015)), and Netherlands (e.g. Cabauw, Schlag et al. (2016)).

When considering the seasonality of eBC, monthly concentrations remained relatively constant (Fig. 5a). This result was somewhat expected since the largest source of eBC at the measurement site was traffic and the traffic counts at the site are relatively constant when excluding specific time-periods such as public holidays and summer vacations (e.g. Luoma et al., 2020). Similar eBC patterns have been observed in other studies performed at traffic sites (Kutzner et al., 2018; Singh et al., 2018).

For estimating the percentage of local eBC emissions, the eBC measured at an urban background site (Kallio) and at a regional background site (Luukki) (Luoma et al., 2020) were subtracted from the eBC measured at the street canyon site, and the remaining eBC were divided by the street canyon eBC concentration (Fig. S8). A median contribution of up to 71 % when subtracting eBC from the urban background site was obtained for local eBC during summer. This percentage could be considered as being a roughly descriptive estimate of the influence of the on-site traffic eBC emissions. When subtracting eBC from the regional background site, a local eBC median percentage of up to 85 % was obtained. The local contribution of eBC was lower during winter, possibly due to stagnant conditions during this period of the year favoring eBC accumulation at the background sites after regional transport. The presence of other sources of eBC (e.g. $BC_{WB}$) can also contribute to the observed results, especially in winter when the background concentrations of eBC are higher than in summer (Luoma et al., 2020).

Interestingly, the highest median concentrations of eBC (Fig. 5a) and local contribution of eBC (Fig. S8) were reached in August. A high concentration of $NO_x$ was also observed in August (Fig. 5b), even though the lowest concentrations were observed in spring and early summer when VOC consumption of NO is higher, the atmospheric dispersion is more efficient in summer, and a decrease in traffic rates during holiday´s period (June–July) is observed. A similar increase in eBC concentrations during summer months has been verified in previous studies conducted in environments dominated by traffic emissions (Healy et al., 2017; Singh et al., 2018). The elevated concentration of eBC in August might be partly explained by poorer dispersion as the ventilation coefficient characterizing atmospheric dilution clearly decreased in this month (Fig. 5c). However, eBC concentrations were also fairly high in June and July, which is surprising considering the diminishing of traffic during holidays.

A factor that might have influenced the measured eBC concentrations is the light absorption enhancement due to the coating of BC particles (e.g. more SOA present during summer for coating and thus for absorption enhancement). This so-called "lensing effect" has been previously described in other studies (e.g. Drinovec et al., 2017; Zhang et al., 2018). As shown in Fig. S9a, the coating factor (CF) slightly increased during spring and summer. Similarly, the $k$, which has been found in previous studies to being applicable as a proxy for the determination of particle´s coating (Drinovec et al., 2017), was also lower in our study during summer (Fig. S9b). Thus, the seasonal variation in both CF and $k$ indicate that the particles were





potentially more coated during the warmest periods of the year. For this reason, it is possible that the lensing effect was more pronounced during summer than during winter. However, additional studies are still needed to confirm this hypothesis.

The BC$_{FF}$ followed the seasonal variation of eBC (Fig. 5d), demonstrating that traffic is the dominant source of eBC at the measurement site. The median biomass burning percentage (BB) was small during most of the seasons but increased up to

about 20 % during winter (Fig. 5e), which suggests an increase of BC$_{WB}$ during the coldest season (Fig. 5f). In Helsinki, wood combustion is typically of residential origin and can be regionally transported into the measurement site (Aurela et al., 2015; Helin et al., 2018; Hellén et al., 2017).

**3.3 Diurnal and weekly variation of atmospheric aerosol particle composition**

Diurnal and weekly profiles of atmospheric aerosol particle constituents can provide insights into sources and atmospheric

processes since diurnal changes are affected by an interplay of factors, including e.g. daytime photochemistry, gas-particle partitioning, and local source emissions.

The diurnal and weekly pattern of BC$_{FF}$ was similar to the one obtained for eBC and NO$_x$ (Figs. 6a–c and Figs. S10a–b), reflecting the dominant contribution of traffic to the obtained profiles. These species had the largest daily variations, with maximums during morning and afternoon rush hours. Similar diurnal cycles of BC$_{FF}$ have been observed in previous studies

conducted at other European sites influenced by traffic (e.g. Crilley et al., 2015; Fuller et al., 2014; Henrich et al., 2011; Jereb et al., 2018). A distinct seasonality on BC$_{FF}$ rush hour concentrations was observed (Fig. S11). In winter, BC$_{FF}$ concentrations were similar during morning and afternoon rush-hour peaks whereas on the other seasons BC$_{FF}$ concentrations were clearly higher during the morning rush-hour. These results suggest that planetary boundary layer dynamics are playing a major role in the measured BC$_{FF}$ concentrations since VC was generally higher during afternoon (Fig. 6d), increasing over the day during

the warmer seasons while remaining constant during wintertime (Fig. S7). The influence of mixing conditions is particularly evident during summer mornings when weak mixing conditions were observed (Fig. S7), causing high concentrations of BC$_{FF}$ even though cold start emissions are insignificant compared to winter and the holiday´s period is included.

The BC$_{WB}$ diurnal and weekly variations were also investigated (Figs. 6e–f and Figs. S10c–d). Since concentrations were low during most of the year (Fig. 5f), only winter concentrations were considered for diurnal variations. The concentrations of

BC$_{WB}$ slightly increased during late night and during weekends, which is likely associated with the use of wood burning for heating purposes. The same evidence was observed by Herich et al. (2011) in an urban background site located in the city center of Zurich.

For a more comprehensive overview, the diurnal and monthly variations of BC$_{FF}$ during workdays (Fig. S12a) were also compared with the corresponding weekend variations (Fig. S12b). Clear differences were observed for these periods of the

week, as expected for a traffic-dominated site (e.g. Helin et al., 2018). Similar diurnal and monthly patterns as for eBC were observed for NO$_x$ during weekdays and weekends (Figs. S12c–d). The previously referred increase in BC$_{WB}$ during weekend´s late evenings is particularly notorious in Figs. S12e–f, especially for the coldest months, which supports the likely influence of wood burning from regional transport during those periods of the year.



In terms of the NR-PM$_1$ constituents, distinct diurnal variations were observed. The concentrations of organics increased only
slightly during the morning rush hour, which suggests a small contribution from traffic induced sources (Fig. S13a). However,
when organics diurnal and monthly variations during workdays (Fig. S12g) are compared with the corresponding weekend
variations (Fig. S12h), a clear seasonality is observed. During winter rush-hours, particulate organics content was clearly
higher during workdays when compared to weekends, revealing the high influence from traffic during this season. In spring
and autumn workdays, organics increased during morning rush-hours due to traffic, since this variation was not observed
during weekends, but decreased during the afternoon rush-hours suggesting a role of meteorology (Figs. S12i–j). During
summer, the variation of organics concentration during workdays and weekends was less clear, which is likely caused by a
dominant influence of biogenic organics formed by oxidation processes during this season of the year. In fact, the organics
variation is relatively similar to the one observed by Heikkinen et al. (2020) at a rural site in Finland.

Nitrate was increasing throughout the night and decreasing during day, following the boundary layer dynamics and temperature
profiles (Fig. S13b). Ammonium and sulfate did not reveal a clear daily variation (Fig. S13c and Fig. S13d). This was expected
since their concentrations were likely affected by the occurrence of long-range or regional transport. Therefore, the low
influence from local traffic density and/or gas-phase production results in the inexistence of daily patterns of these atmospheric
constituents at the measurement site.

### 3.4 Seasonal variations of aerosol particle number and mass size distributions

In addition to PM$_1$ variations in chemical composition, the seasonal variations in particle number and mass size distributions
observed during our long-term measurements were evaluated in more detail. As can be seen in Fig. 7a, the median particle
number size distributions showed a clear seasonal variability. The wintertime particle number size distribution was
characterized by a dominating nucleation mode that peaked at ~11 nm. Particle size distributions consisting of a major peak at
5–12 nm have been associated to fresh traffic emissions (Al-Dabbous and Kumar, 2015; Rönkkö and Timonen, 2019).
On the other seasons, the particle number size distributions had two modes, the first one appearing at a size close to 11 nm and
the second mode occurring in the Aitken mode range with a maximum between 40–60 nm. Similar size distributions have been
found in other studies performed at traffic environments (Barone and Zhu, 2008; Ondráček et al., 2011; Rönkkö et al., 2017;
Virtanen et al., 2006; Voigtländer et al., 2006). As referred previously, the mode at ~10 nm has been related to fresh traffic
emissions, including both direct emissions of nanoparticles from the exhaust and particles formed through nucleation of low-
volatile vapors after their cooling; while the second mode has been associated to coagulation of these particles and/or
condensation of low-volatile vapours on the primary particles (Barone and Zhu, 2008; Ondráček et al., 2011) even though soot
might have also played a role. Interestingly, the second mode was clearly higher in our study during summer comparatively
to the first mode. This result suggests that SOA formation, fueled by large emissions of SOA precursors and the presence of
reactive tracer atmospheric oxidants, prevails during this time of the year. Elevated summer concentrations have been also
observed in lung deposit surface area (LDSA) results from recent measurements performed in Helsinki (Kuula et al., 2020).



Furthermore, Rivas et al. (2020) has found that biogenic emissions were an important contributor to ultrafine particles in Helsinki during summer, both in urban background and street canyon environments.

The seasonality of particle mass size distributions was as well investigated (Fig. 7b). Similar distributions were observed for the different seasons, with a maximum mass concentration occurring at around ~258 nm. The mass mode was slightly higher during summer and larger in summer and spring comparatively to the other seasons.

The diurnal variation of particle number size distributions was also evaluated (Fig. 8). The size distributions of particles observed during morning and afternoon rush hours varied greatly between seasons. From autumn to spring, the maximum of number size distributions ranged between 7 and 40 nm, while in summer high concentrations were observed up to ~80 nm likely due to the presence of SOA. In winter, number size distributions were comparable during morning and afternoon rush-hours. However, in the other seasons, maximums of the number size distributions were much lower during afternoon rush-hours due to the previously referred VC effect. This effect seems to impact mostly particles with larger sizes, suggesting that planetary boundary layer and meteorological conditions play an important role in size distributions.

## 3.5 Effects of local and long-range transport pollution episodes on the measured aerosol particle composition

Air quality in Nordic cities is affected by local and long-range transport pollution episodes. In this study, the influence of both pollution episodes was evaluated. Figure 9 shows a time period when a long-range transport episode caused a long-lasting increase in $PM_1$ (8th to 19th of February 2018). Additionally, a local pollution episode took place after the long-range transport episode (21st to 23rd of February 2018) as a result of an atmospheric inversion. The selected period occurred during winter, when local pollution is particularly severe due to an interplay of factors that include stagnant conditions and generally low mixing height, which favors the local confinement and accumulation of PM (e.g. Brown et al., 2006). During both pollution events, the hourly median levels of $PM_{2.5}$ reached/exceed the concentration of 25 µg m$^{-3}$, which is the guideline limit given by the World Health Organization for the 24-hour mean $PM_{2.5}$ mass (WHO, 2006).

As observed in Fig. S14a, south-eastern to south-western winds brought polluted air-masses to Helsinki during the long-range transport event. This is typically observed for long-range transport episodes detected in Helsinki (Niemi et al., 2009; Timonen et al., 2013). The pollution event caused an increase in $PM_{2.5}$ hourly median concentration of up to 24.9 µg m$^{-3}$, with $PM_1$ concentrations reaching up to 22.6 µg m$^{-3}$. The median relative contribution of inorganic species to $PM_1$ was highest, accounting for 52 %, while the contributions of organics and eBC were 39 % and 9 %, respectively. Of inorganic species, sulfate dominated $PM_1$ with a median fraction of 25 %, suggesting that the long-range transport particles were formed from coal-fired plants and industrial sources. Nitrate and ammonium contributed to 16 % and 11 % of $PM_1$ respectively. The long-range transport aerosols caused mostly an increase in mass concentrations within the accumulation mode size range (approx. 200–800 nm). According to back-trajectories, this long-range transport was originated from eastern or eastern/central Europe (i.e. Russia, Belarus, Poland and Baltic countries).

In contrast, an increase in the contribution of traffic-related particulate matter was clearly observed during the local pollution episode under calm wind conditions. The hourly median $PM_1$ and $PM_{2.5}$ levels reached 23.5 µg m$^{-3}$ and 28.6 µg m$^{-3}$ respectively,



with a dominant fraction arising from the organics (up to 58 %) and eBC (up to 42 %). A simultaneous increase in $NO_x$ levels

was observed. Back trajectories revealed that Nordic air masses were dominant during the local pollution event (Fig. S14b), which justifies the low concentration of inorganics during the event. Contrary to the long-range transport episode, the particle number concentrations during the local pollution episode increased sharply, especially in nanoparticle size with diameter <100 nm. The impact of traffic-related particles on health is related to their small size, since they can penetrate efficiently the respiratory tract (e.g. Carvalho et al., 2011). Therefore, considering the morbidity and lethality associated with carbonaceous

aerosols (e.g. Lelieveld et al., 2015), the observed increase in ultrafine particles during the local pollution event is of particular significance.

Besides the case study showed above, we also investigated the role of long-range transport and local pollution over all measurement period (2015–2019). The $PM_{2.5}$ levels were used for that purpose by determining the median contribution of $PM_1$ chemical constituents for different $PM_{2.5}$ mass classes (Fig. 10). A similar approach was used by Petit et al. (2015) but by using

a mass classification based on $PM_1$. The unanalyzed fraction of $PM_{2.5}$ determined by subtracting the sum of total NR-$PM_1$ and eBC from the $PM_{2.5}$ was included in our study. By frequency, most of the high pollution episodes (e.g. $PM_{2.5}$ bins > 20 µg m$^{-3}$ that includes 2.3% of measurement time) occurred during autumn, winter and early-spring (see lower diagram in Fig. 10 and Figs. S15 and S16). These episodes of cool season were often characterized by increased inorganics fraction, especially the fraction of sulfate, emphasizing the role of long-range transport and fossil fuels as major sources (Fig. S16). On the other hand,

organics were higher during spring and summertime episodes (Fig. S16), which can be associated to SOA formation. The unanalyzed fraction of $PM_{2.5}$ was highest during the spring-time street dust periods.

In general, the fraction of organics was higher for $PM_{2.5}$ concentrations of up to 20 µg m$^{-3}$ (top diagram in Fig. 10). However, the contribution of organics decreased from 36.7 % to 28.3 % with the increase in $PM_{2.5}$. On the other hand, the fraction of nitrate continuously increased from 6.4 % to 9.4 % for concentrations of up to 30 µg m$^{-3}$, while sulfate and ammonium

contribution increased from 8.6 % to 9.9 % and from 3.9 % to 4.6 % respectively for $PM_{2.5}$ of 5 to 25 µg m$^{-3}$. The relative contribution of $BC_{FF}$ remained relatively constant over the different $PM_{2.5}$ masses, varying between 7.8 % and 9.3 %. The almost constant $BC_{FF}$ contribution indicates that the local primary emissions from traffic exhaust are not usually the main source of high $PM_{2.5}$ episodes, since primary exhaust emissions are characterized with the high fraction of $BC_{FF}$ (Fig. 9). The contribution of $BC_{WB}$ was the smallest for all $PM_{2.5}$ concentration bins.

Interestingly, the unanalyzed fraction of $PM_{2.5}$ increased from 10 to over 30 µg m$^{-3}$ of $PM_{2.5}$, reaching a contribution of 44.7 % at the latest concentration bin. Since the contribution of inorganics decreased during the highest concentrations of $PM_{2.5}$, regional/local pollution episodes (e.g. biomass burning and street dust) are likely contributing to the high unanalyzed $PM_{2.5}$ fraction. As observed in Fig. 10, the ratio of $BC_{WB}$ to eBC also increased for the highest levels of $PM_{2.5}$, suggesting that biomass burning (e.g. larger particles caused by wildfires) is playing a role on the $PM_{2.5}$ unanalyzed fraction. Furthermore, the

$PM_{10}/PM_{2.5}$ ratio increased for $PM_{2.5}$ concentrations over 20 µg m$^{-3}$, with the increase being particularly significant for $PM_{2.5}$ levels over 30 µg m$^{-3}$. These results suggest that both biomass burning and street-dust pollution episodes are contributing to





the increased unanalyzed fraction of PM$_{2.5}$, with street dust air pollution possibly being the most important contributor due to its prominence during spring.

## 4 Conclusions

In this study, highly time-resolved measurements were performed at a street canyon for a period of four and half years. The results revealed the seasonal and daily variances of PM$_1$ chemical constituents that influence local air quality in Helsinki. Generally, organics had the highest contribution to PM$_1$ mass, followed by eBC and inorganic species (sulfate, nitrate and ammonium). A clear seasonal variation was observed for organics, with the highest concentrations being measured during summer. The organics formed from traffic seemed to be more prominent during the coldest months, while biogenic organics

likely prevailed during summer. Inorganic species presented similar diurnal cycles that were mainly driven by the dynamics of the planetary boundary layer and long-range transport. The BC$_{FF}$ remained mostly constant over the seasons, reflecting the relatively regular traffic volume, but had the highest diurnal and weekday to weekend variability. The BC$_{WB}$ was higher during weekend´s late evenings in winter. The trend analysis revelated that BC$_{FF}$ and organics are decreasing over the years, while only a minor decrease in BC$_{WB}$ was observed. Regarding measured PM$_1$ inorganic species, a decreasing trend was observed

for nitrate but not for sulfate and ammonium. Particle number and mass size distributions in different seasons revealed the possible influence of SOA during summertime, while ultrafine traffic particle production and exposure was predominant during wintertime with possible consequences for public health.

The investigation of pollution episodes showed that both local and long-range transported pollutants can cause elevated PM$_1$ and PM$_{2.5}$ concentrations. A large fraction of PM$_{2.5}$ consisted of inorganics during long-range transport episodes, while local

episodes showed an increase in eBC and organics. Ultrafine particle number concentration was particularly significant during local episodes, while particle mass was higher during long-range transport. Street dust seemed to be an important contributor for the PM$_{2.5}$ missing fraction in spring. Even though the use of studded winter tyres and the spreading of traction sand to sidewalks and streets are common measures to increase traffic safety in Finland during winter, the influence of street dust on air pollution might still be significant, which would require additional efforts to prevent dust particles from stirring up into

the atmosphere. Altogether, our results show that measures tackling the main constituents of the submicron aerosol fraction require both national-based policies to reduce local pollutants and concerted actions at European level for efficiently complying with PM$_{2.5}$ legislations and guidelines.

The primary emissions from traffic are expected to be under continuous change. The urgent need for greener sources of energy will shift fossil fuels towards other type of energy supplies at a fast pace, and the development of more efficient catalysts and

cleaner technologies is on-going. Due to the stricter emission laws and improvements on vehicle technologies, a decrease in the contribution of traffic to PM$_1$ is expected to occur. Even though the traffic-related PM$_1$ emissions will likely reduce, continuous long-term measurements at urban environments will be of paramount importance, specially concerning non-exhaust emissions that will be of growing concern during the transition from fossil fuel vehicles. Long-term studies using high-time



resolution instrumentation are still required to understand the impact of atmospheric pollutants on air quality in traffic
environments, which is fundamental for the development and implementation of effective pollution mitigation strategies by
policymakers in future air quality and climate decisions and to protect human health.

*Data availability.* The data shown in the paper is available on request from corresponding author.

*Author contribution.* HT, TR, JVN designed the experiments and MA, KT, JVN, HP, AK, LP, LB carried them out. LB, MF,
HT, MA, SS, AH, LK, HP, AK contributed to the data analysis and interpretation. HT, TR, LP, JVN, SS contributed to the
funding acquisition and supervision. LB wrote the manuscript. All authors participated to the interpretation of the results and
paper editing.

*Competing interests.* The authors declare that they have no conflict of interest.

*Acknowledgements.* We acknowledge funding from Black Carbon Footprint project funded by Business Finland and
participating companies (Grant 528/31/2019), from European Regional Development Fund, Urban innovative actions initiative
(HOPE; Healthy Outdoor Premises for Everyone, project nro: UIA03-240) and from MegaSense Growth Engine: Air Quality
Monitoring funded by Business Finland (Grant 7517/31/2018). Long-term research co-operation and support from HSY to this
project is gratefully acknowledged. The authors gratefully acknowledge the NOAA Air Resources Laboratory (ARL) for the
provision of the HYSPLIT transport and dispersion model and/or READY website (https://www.ready.noaa.gov) used in this
publication.

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





**Table 1: The concentrations of PM$_1$ constituents measured by ACSM (10.06.15–31.12.19) and MAAP (01.06.15–31.12.19), and of PM$_{2.5}$ measured by TEOM (01.06.15–31.12.19). NR-PM$_1$ was obtained by summing all the constituents measured by the ACSM, while PM$_1$ is the sum of NR-PM$_1$ and eBC from MAAP. The eBC from fossil fuel combustion (BC$_{FF}$) and from wood burning (BC$_{WB}$) measured by aethalometer (30.11.15–31.12.19) are also included. Both hourly mean (± standard deviation), median (± median absolute deviation), 25$^{th}$ and 75$^{th}$ percentiles, and maximum concentrations are included.**

| Analytes | Mean ($\mu g\ m^{-3}$) | Median ($\mu g\ m^{-3}$) | 25th percentiles | 75th percentiles | Max ($\mu g\ m^{-3}$) |
|---|---|---|---|---|---|
| Org | 2.89 ± 2.07 | 2.34 ± 1.12 | 1.40 | 3.84 | 24.40 |
| SO$_4$ | 0.67 ± 0.65 | 0.50 ± 0.29 | 0.24 | 0.86 | 7.44 |
| NH$_4$ | 0.33 ± 0.38 | 0.22 ± 0.15 | 0.10 | 0.43 | 3.91 |
| NO$_3$ | 0.56 ± 0.73 | 0.31 ± 0.20 | 0.15 | 0.66 | 9.61 |
| eBC | 0.97 ± 0.93 | 0.71 ± 0.40 | 0.37 | 1.27 | 14.68 |
| BC$_{FF}$ | 0.68 ± 0.66 | 0.48 ± 0.29 | 0.24 | 0.90 | 11.90 |
| BC$_{WB}$ | 0.13 ± 0.16 | 0.08 ± 0.05 | 0.04 | 0.15 | 3.73 |
| NR-PM$_1$ | 4.47 ± 3.16 | 3.64 ± 1.64 | 2.24 | 5.82 | 29.96 |
| PM$_1$ | 5.49 ± 3.67 | 4.59 ± 2.00 | 2.86 | 7.16 | 30.52 |
| PM$_{2.5}$ | 7.20 ± 4.76 | 6.25 ± 2.45 | 4.10 | 9.20 | 96.35 |





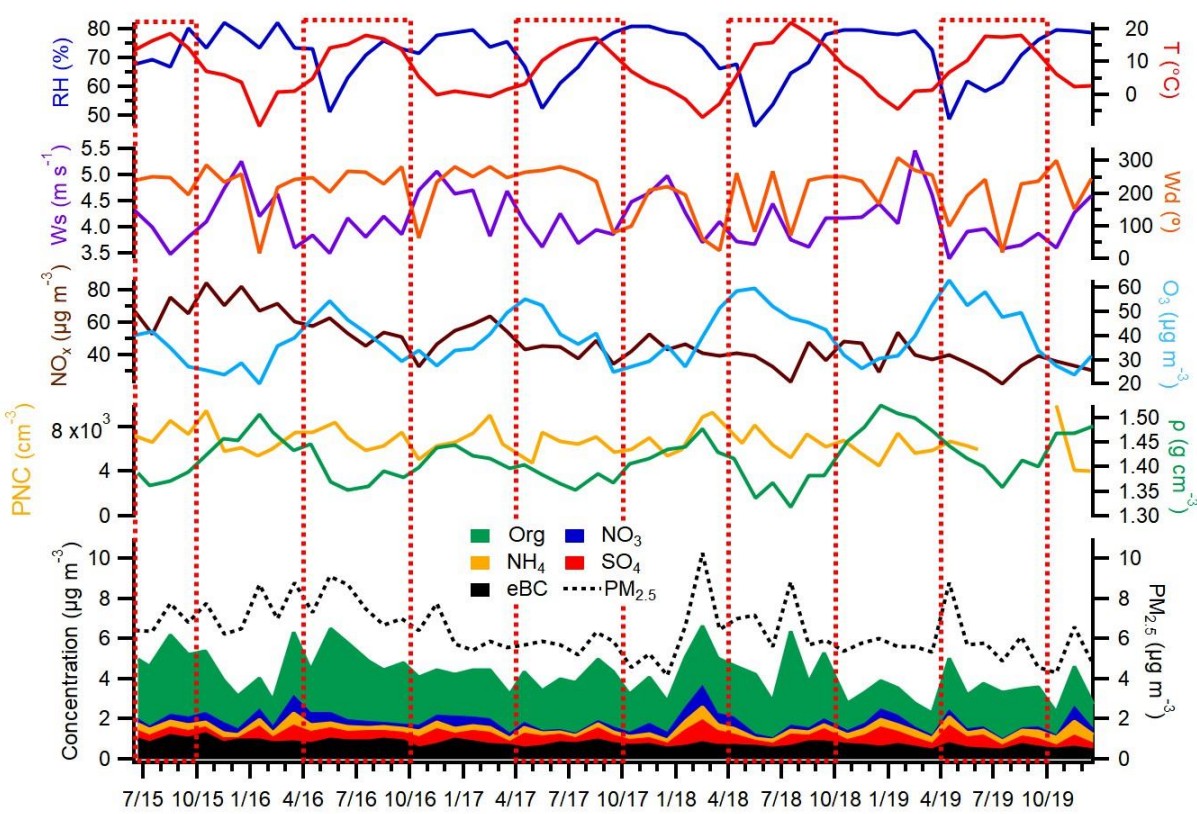


**Figure 1: Temporal variation (monthly median) of PM$_1$ chemical species (organics, NO$_3$, NH$_4$, SO$_4$ and eBC), PM$_{2.5}$ (determined by TEOM), PNC (measured by DMPS), particle density, atmospheric oxidants (NO$_x$ and O$_3$) and the most relevant meteorological parameters (Wd (monthly mean), Ws (monthly mean), T, RH). Red dotted rectangles represent the warmest period of the year (Apr–Sep).**




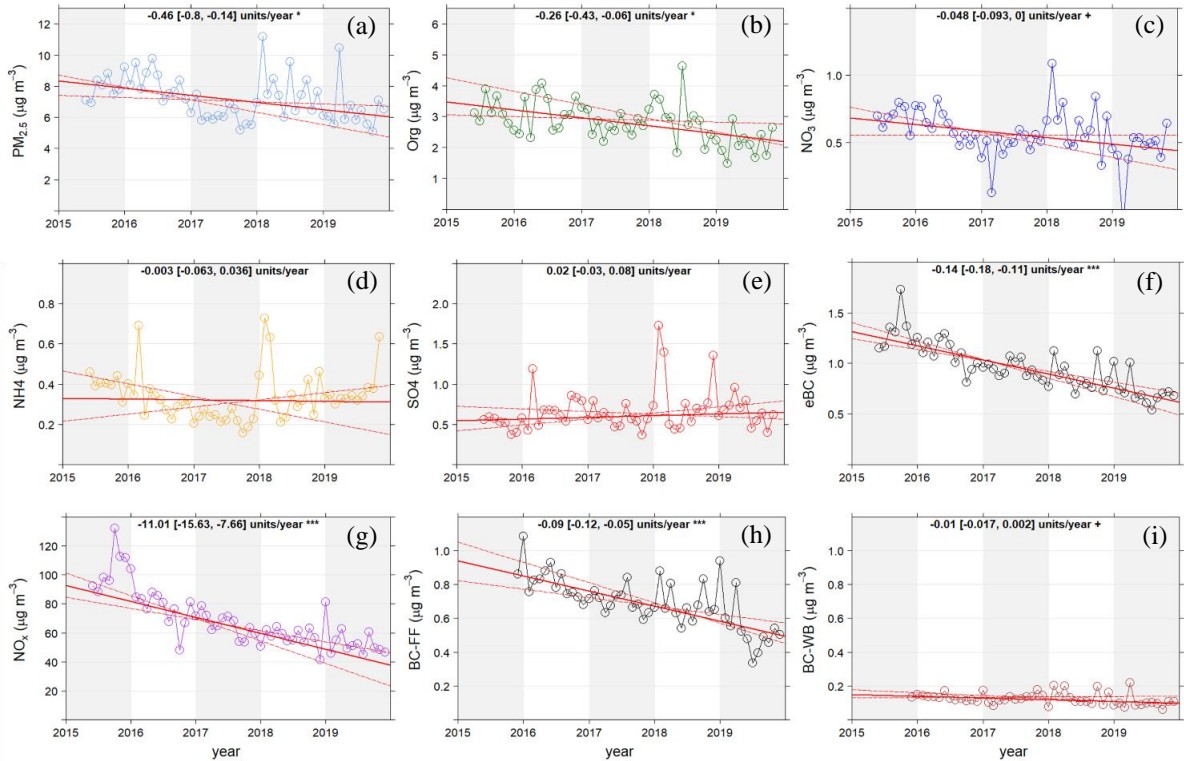

**Figure 2: Trends analysis results of selected pollutants (a: PM$_{2.5}$, b: Organics, c: NO$_3$, d: NH$_4$, e: SO$_4$, f: eBC, g: NO$_x$, h: BC$_{FF}$, i: BC$_{WB}$. The plots show the deseasonalized monthly mean concentrations of different pollutants. The solid red line shows the trend estimate and the dashed redlines show the 95 % confidence intervals for the trend. The overall trend value is presented at the top center of each plot as corresponding units per year and the 95 % confidence intervals in the slope are presented in the brackets. The symbols shown next to each trend estimate indicate the statistical significance of the trend estimate (p < 0.001 = ∗ ∗ ∗, p < 0.05 = ∗ and p < 0.1 = +).**






**Figure 3: Time series of the PM$_1$ species monthly medians mass fractions measured by ACSM (Org, SO$_4$, NO$_3$, NH$_4$) and MAAP (eBC). Pie-plots represent the relative contribution of each species during different seasons of the year (spring (Mar–May), summer (Jun–Aug), autumn (Sep–Nov) and winter (Dec–Feb)).**





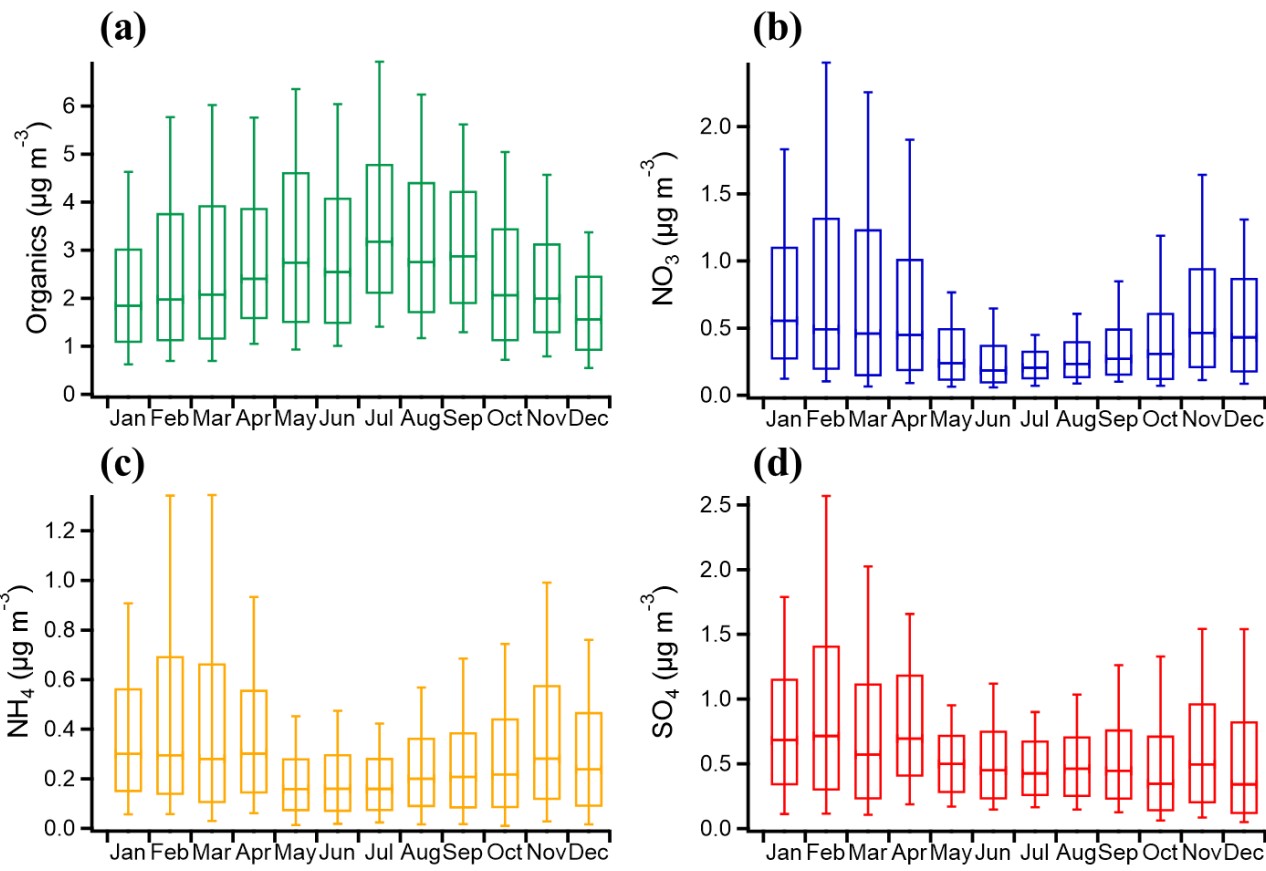

Figure 4: Monthly variation of the NR-PM$_1$ constituents (organics, NO$_3$, NH$_4$ and SO$_4$) measured by ACSM. In each box, the mid-line shows the median value for each x-value component, whisker bottom and top correspond to the 10$^{th}$ and 90$^{th}$ percentiles, and box top and bottom correspond to 75$^{th}$ and 25$^{th}$ percentiles respectively.






**Figure 5: Monthly variation of eBC, NOₓ, VC, BC_FF, BB and BC_WB. In each box, the mid-line shows the median value for each x-value component, whisker bottom and top correspond to the 10th and 90th percentiles, and box top and bottom correspond to 75th and 25th percentiles respectively.**





**Figure 6: Diurnal variation of eBC, NO$_x$, BC$_{FF}$, VC, BB and BC$_{WB}$. In each box, the mid-line shows the median value for each x-value, whisker bottom and top correspond to the 10$^{th}$ and 90$^{th}$ percentiles, and box top and bottom correspond to 75$^{th}$ and 25$^{th}$ percentiles respectively. Note that BB and BC$_{WB}$ diurnal variations include only winter months (Dec–Feb).**






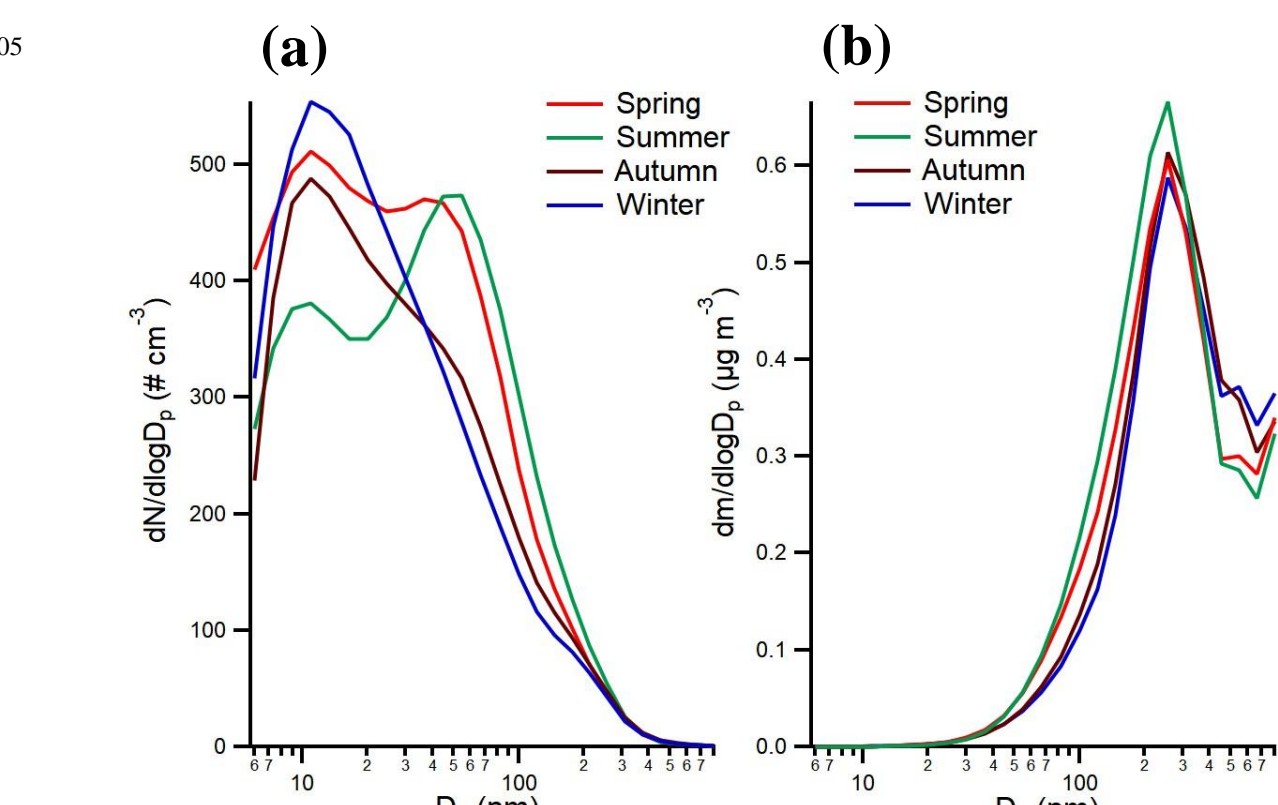

**Figure 7: Seasonal variation of median particle number and mass size distributions (note that $D_p$ corresponds to the particle´s mobility diameter).**





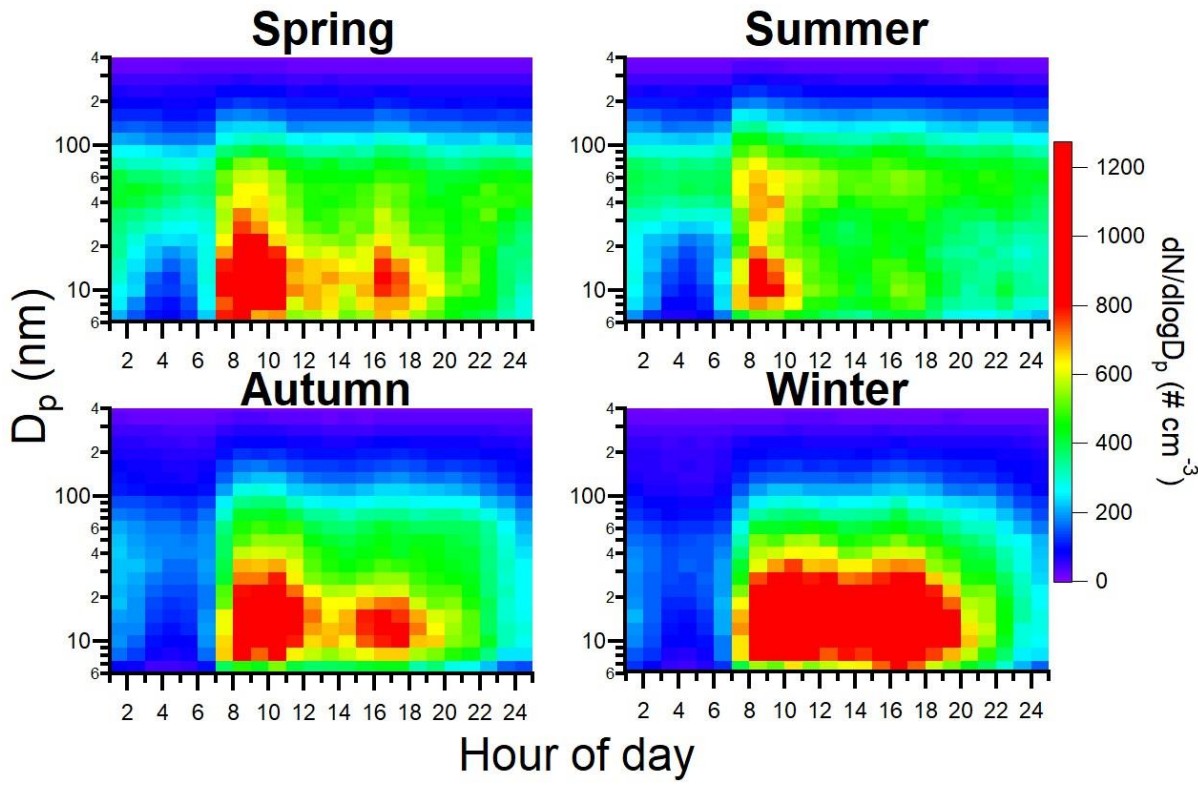


**Figure 8: Hourly variation of median particle number size distributions over the different seasons of the year (note that $D_p$ corresponds to the particle´s mobility diameter).**

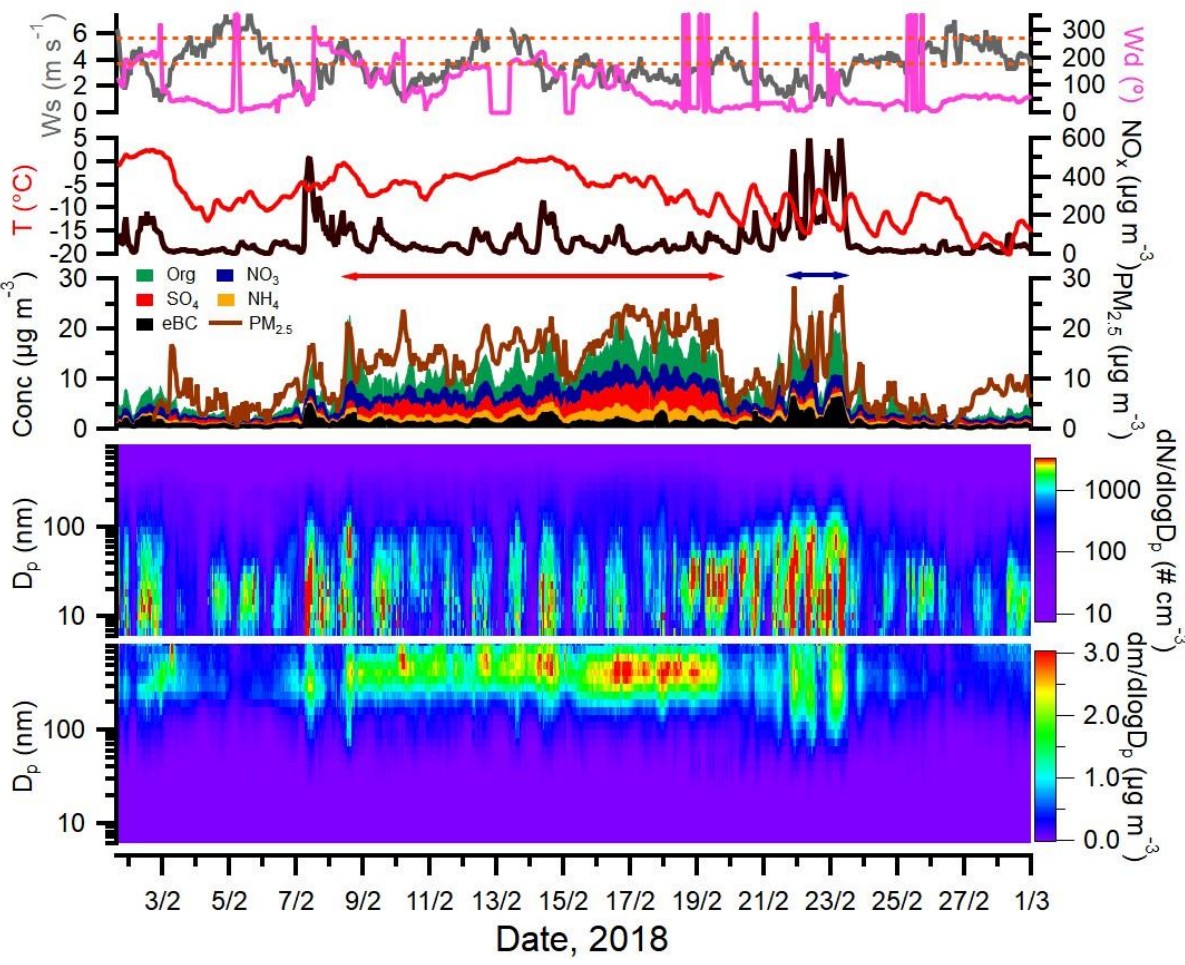

**Figure 9: Hourly median PM₁ composition and concentration, PM₂.₅ concentration, mass and number size distributions, and other atmospheric relevant parameters (T, NOₓ, Ws (hourly mean), Wd (hourly mean)) during a case study where local and long-range transport pollution events occurred. The red sideway arrow indicates the time period of the long-range transport event, while blue sideway arrow indicates the local pollution episode. Orange lines represent south and west wind directions (180° and 270°).**





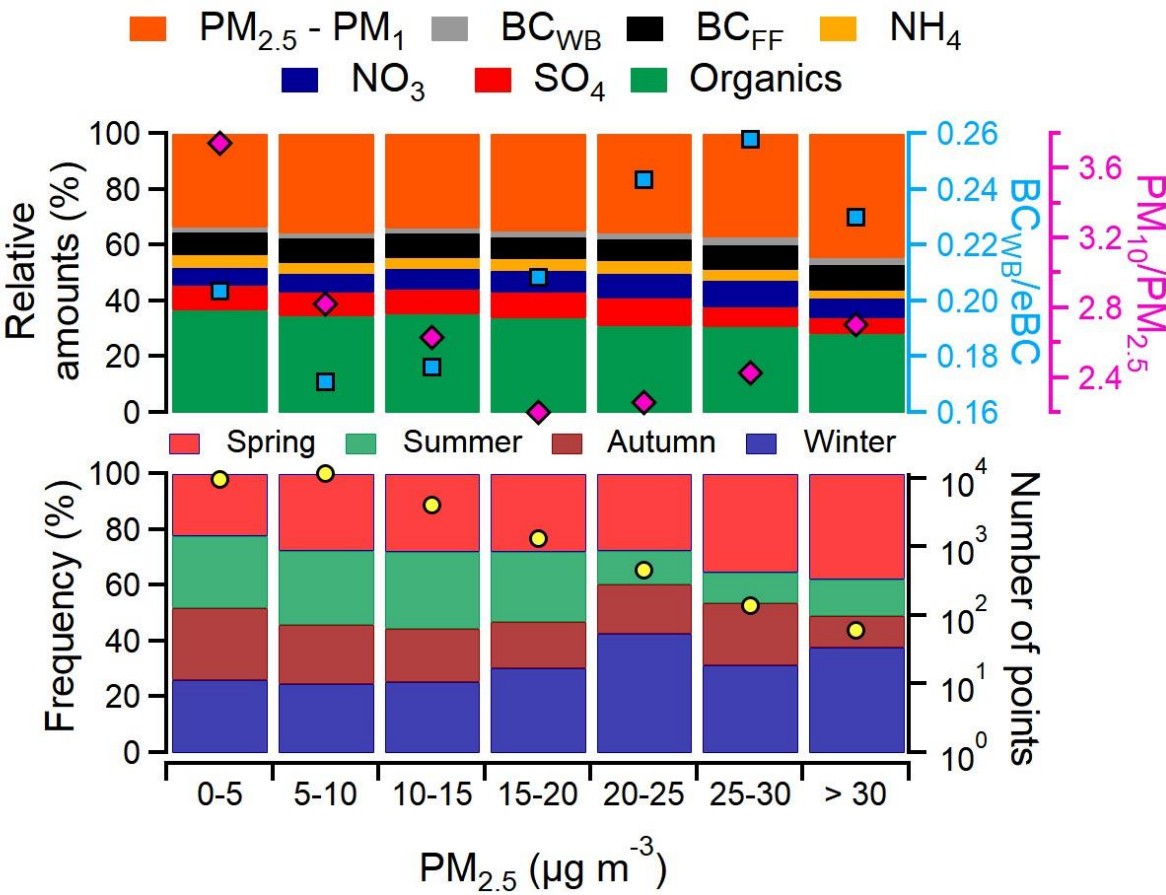

**Figure 10: Median relative contribution of chemical composition to different PM2.5 concentration bins (top) and seasonal frequency, including number of hour points (yellow dots), in each PM2.5 concentration bin (bottom). The ratios between BCWB to eBC (blue squares) and PM10/PM2.5 (purple diamonds) were also included.**