# Peer review of "In-depth characterization of submicron particulate matter interannual variations at a street canyon site in Northern Europe"

_Atmospheric Chemistry and Physics, 2020_

## Referee Comment (RC1) · Anonymous Referee #1 · 25 Jan 2021

The paper by Barreira et al. summarizes measurements of aerosol composition and total mass concentration and meteorological parameters at an urban site in Helsinki during 2015-2019. Measurements include PM1 from an ACSM and MAAP, size distributions and absorption by an aethalometer, as well as PM2.5 mass concentration from TEOM. They conclude that long range transport and local pollution along with boundary layer height and dilution extent control aerosol mass concentrations at the site. During long-range transport pollution episodes, PM1 was dominated by inorganic species, mainly sulfate while during local pollution episodes (typically in winter), organics and BC were significant. Wood burning contribution to BC and optical absorption was observed during winter, and especially at night. Seasonally, organic concentra-

tions increased during summer, suggesting contribution of SOA. Trend analysis indicate that during the measurement period, there was a decrease in organics, nitrate, and BC concentrations. The paper is overall well-written and interesting, but the scope of the paper fits better with a Measurement Report since there wasn't any new insight on general atmospheric sources of aerosols or aerosol formation and this is not the first study of aerosol measurements in Helsinki. With the following revisions, I recommend publishing it as a Measurement Report (but not research article).

L91. Clarify that the provided size range is in dva. L100. (NH4)2SO4 is ammonium sulfate. L116-117. These two sentences where confusing. If measurements were done at 1-min intervals, the DL should also be quoted for 1-min averaging times and not 10 min and vice versa. L150-155. It was confusing as to what density was used to convert dm to dva. One sentence indicates a constant 1.5 g/cc. This is followed by the composition-dependent equation for density. Also, why was such a low value for organics density used? Especially during long-range transport, OA density is higher than 1.2 because of dominance of oxygenated species. L158-159. Why wasn't a transmission efficiency applied to the volume distributions from SMPS to really count the particles in the ACSM range? Dm=549 nm is ~dva=780 which is larger than 50% cut-size of the ACSM; therefore I don't think the comparisons between ACSM and SMPS are correct. More concerning is that the bounce correction was determined by this comparison. L180-185. This is also related to the point above. The mass concentration calculations for ACSM seem circular. If the first density estimate used to convert SMPS volume to mass is based on an equation which uses mass estimates of ACSM and if those mass estimates relative to BC are not correct, then the estimated density and SMPS mass concentration are not correct, so the ratio of (SMPS-BC) mass to ACSM mass is not correct. Please explain why you think this calculation is correct. L248. It's mentioned that the 4.5 yr dataset might not be long enough for trend analysis. How far back are similar data available? It seems some measurements are available since 2013. Can these two datasets be combined for just the trend analysis? L307. The seasonal explanation of VC doesn't match the monthly behavior as shown

in Fig. 5. It appears that VC is high during summer and low in winter. Why do authors say that ventilation was low in August? Add VC after 'ventilation coefficient' to define it since it's only discussed in SI. L308. So what's the source of high BC during holiday season/summer? Long range transport? L337. Cold start emissions during summer are still important although the duration of such conditions might be shorter. L348. Why can't the increased BCWB be from local sources? L385. Too qualitative of a statement. Please indicate a number (either exact or say larger by ##%). L400. Change exceed to exceeded L413. What was the concentration of BCff during this local pollution event? In the long-range transport case discussed in the following paragraph, it is indicated that during the local events relative contribution of BCff is not higher than during the long-range transport times and I find that surprising. Is that because its mass conc. is low relative to all the secondary species so the relative contribution stays more or less the same?

L 446. I think it makes more sense to look at the PM2.5 fraction of PM10 rather than the ratio of PM10/PM2.5. Why was the ratio used in the analysis?

---

## Referee Comment (RC2) · Anonymous Referee #2 · 25 Jan 2021

Overview: The manuscript by Barreira et al. used four major instruments (ACSM, DMPS, AETH, MAAP) to measure the chemical composition, diurnal trend, seasonal trend, and simple source apportionment of PM1 and PM2.5 from an urban street canyon in Helsinki, Finland, for four and half years. The results demonstrate that various kinds of chemical composition have been decreasing during the measurement time, and that season trend of the organic, black carbon, and particle size distributions were also described in this study. Overall the study is clearly written and easy to follow. I recommend for publication after the following points addressed.

Major Comments: In section 3.5, the author discussed the effects of local and long-

range transport by comparing a few factors for defining long range transport vs local formation. The author also used satellite data to support these arguments. I am curious whether the author did any examination based on certain tracer ions from previous studies to perform a more detailed the source apportionment of the data collected. For instance, have the authors analyzed m/z 82 signal to examine the fraction of the IEPOX-SOA? (Budisulistiorini et al., 2013, Hu et al., 2016), or m/z 60 to understand the concentration of levoglucosan or biomass burning (Bougiatioti et al., 2014)). With four and a half year data, the author should probably also use PMF to analyze the data and look for any information that the PMF may be able to provide. For instance, the author can look at the ratio between more aged organic components vs less oxidized, which may help distinguish aerosols from long range or local transportation.

It was a bit confusing when the author described the density conversion in line 150-155. For instance, the author described a constant density of 1.5 g cm-3 was used to convert mobility diameter to vacuum aerodynamic diameter, without specifying the reference. Then the author calculated the gravimetric density to be 1.42 g cm-3. Why would the author not use 1.42 g cm-3 to reconvert the mobility diameter to vacuum aerodynamic diameter again to make the results more accurate?

Minor Comment: L 135: Please define BC(FF) and BC(WB) when it first appear.

References:

Budisulistiorini, S. H.; Canagaratna, M. R.; Croteau, P. L.; Marth, W. J.; Baumann, K.; Edgerton, E. S.; Shaw, S. L.; Knipping, E. M.; Worsnop, D. R.; Jayne, J. T.; Gold, A.; Surratt, J. D., Real-Time Continuous Characterization of Secondary Organic Aerosol Derived from Isoprene Epoxydiols in Downtown Atlanta, Georgia, Using the Aerodyne Aerosol Chemical Speciation Monitor. Environ. Sci. Technol. 2013, 47 (11), 5686-5694.

Bougiatioti, A.; Stavroulas, I.; Kostenidou, E.; Zarmpas, P.; Theodosi, C.; Kouvarakis, G.; Canonaco, F.; Prévôt, A. S. H.; Nenes, A.; Pandis, S. N.; Mihalopoulos, N., Pronone

cessing of biomass-burning aerosol in the eastern Mediterranean during summertime. Atmos. Chem. Phys. 2014, 14 (9), 4793-4807.

Hu, W.; Palm, B. B.; Day, D. A.; Campuzano-Jost, P.; Krechmer, J. E.; Peng, Z.; de Sá, S. S.; Martin, S. T.; Alexander, M. L.; Baumann, K.; Hacker, L.; Kiendler-Scharr, A.; Koss, A. R.; de Gouw, J. A.; Goldstein, A. H.; Seco, R.; Sjostedt, S. J.; Park, J. H.; Guenther, A. B.; Kim, S.; Canonaco, F.; Prévôt, A. S. H.; Brune, W. H.; Jimenez, J. L., Volatility and lifetime against OH heterogeneous reaction of ambient isoprene-epoxydiols-derived secondary organic aerosol (IEPOX-SOA). Atmos. Chem. Phys. 2016, 16 (18), 11563-11580.

---

## Author Comment (AC1) · 2 Mar 2021

The referees' comments are addressed in the attached file. The manuscript has been revised accordingly.

Please also note the supplement to this comment:
https://acp.copernicus.org/preprints/acp-2020-908/acp-2020-908-AC1-supplement.pdf

---

## Author Response (AR1)

Response to reviewer comments for manuscript: "In-depth characterization of submicron particulate matter inter-annual variations at a street canyon site in Northern Europe"

Barreira et. al

We thank the reviewers for their constructive comments regarding the paper. Below we address the specific issues point by point. The reviewer's comments are in black and our answers are in blue. Changes to the Manuscript or Supplement Information are highlighted in red (note that the lines referred in reviewer´s comments correspond to the ones in the previous version of the manuscript, while the lines in our answers and in the modified text refer to the updated version).

**Reviewer 1:**

The paper is overall well-written and interesting, but the scope of the paper fits better with a Measurement Report since there wasn't any new insight on general atmospheric sources of aerosols or aerosol formation and this is not the first study of aerosol measurements in Helsinki. With the following revisions, I recommend publishing it as a Measurement Report (but not research article).

The results presented in the manuscript provide recent information on atmospheric pollutants concentrations and give insight about their atmospheric sources, formation and effects of PM mitigation policies implemented during the last years in Northern Europe. Long-term comprehensive studies of atmospheric composition and PM sources are currently scarce, particularly at traffic sites. To our knowledge, this is the first time that a long-term comprehensive characterization of $PM_1$ concentration and composition, including non-refractory $PM_1$ (organics, sulfate, ammonium and nitrate) and refractory $PM_1$ (eBC) constituents, has been performed at an urban street canyon in Northern Europe. Most of these results could only have been achieved through long-term and high time-resolved measurements such as the ones in this study since they require a statistically relevant amount of data. Therefore, these results contribute to the general understanding of atmospheric chemistry and physics and for that reason we think that the presented work would fit into the scope of a research article. Additionally, this manuscript does not just report the concentrations and diurnal/seasonal/annual variations of atmospheric constituents but also discusses scientifically

relevant topics such as the influence of the lensing effect on the measured eBC concentrations and of the atmospheric mixing conditions on the concentration of chemical species. Moreover, the number and mass size distributions measured with DMPS were connected to the discussion on the particle sources and atmospheric processing at the street canyon.

L91. Clarify that the provided size range is in $D_{va}$.

The text was modified as:

P3. L92. The typical 50 % transmission efficiency range in vacuum aerodynamic diameter ($D_{va}$) of the lens is ca. 90-650 nm (Liu et al., 2007).

L100. (NH4)2SO4 is ammonium sulfate.

The text was modified as suggested.

L116-117. These two sentences where confusing. If measurements were done at 1-min intervals, the DL should also be quoted for 1-min averaging times and not 10 min and vice versa.

The MAAP measurements were performed with a 1-min time resolution. However, as mentioned further in the manuscript, hourly mean or median values were used for the data analysis (L175). The LOD for the 1-h time resolution was not determined but it is expected to be lower or equal to the one mentioned for the 10-min span as averaging (or median) reduces the significance of instrumental noise. Since we have not used 10-min averaging, we agree that it is confusing to mention the LOD for that time resolution and the sentence was removed from the text.

L150-155. It was confusing as to what density was used to convert $D_m$ to $D_{va}$. One sentence indicates a constant 1.5 g/cc. This is followed by the composition-dependent equation for density. Also, why was such a low value for organics density used? Especially during long-range transport, OA density is higher than 1.2 because of dominance of oxygenated species.

The particle density used to convert $D_m$ to $D_{va}$ was 1.42 g cm$^{-3}$ throughout the manuscript (median value obtained in this study when employing a composition-dependent equation for density). In terms of OA density, the equation uses a fixed OA density of 1.2 g cm$^{-3}$, which indeed can differ over time

due to changes in the composition as verified during long-range transport episodes. An experimental value for OA density was not possible to obtain with the instrumental setup used in this study. These uncertainties are further discussed in Sect. 3.1 (L228-231). A constant density of 1.5 g cm$^{-3}$ was used in the first version of the manuscript, but it was changed to 1.42 g cm$^{-3}$, in order to avoid confusion, before the manuscript was accepted to ACPD. The text was also modified to:

P6. L160. The upper $D_m$ size range value corresponds to a $D_{va}$ of ~780 nm when employing a density of 1.42 g cm$^{-3}$.

L158-159. Why wasn't a transmission efficiency applied to the volume distributions from SMPS to really count the particles in the ACSM range? $D_m$ =549 nm is ~ $D_{va}$=780 which is larger than 50% cut-size of the ACSM; therefore I don't think the comparisons between ACSM and SMPS are correct. More concerning is that the bounce correction was determined by this comparison.

The DMPS measures a total of 26 size bins from 6 to 801 nm ($D_m$). A total of 23 size bins were selected in this study (up to 549 nm in mobility diameter). As pointed out by the referee, this corresponds the aerodynamic size of ~780 nm that is larger than the 50 % transmission efficiency of the ACSM. One smaller size bin was 454 nm (mobility diameter) that corresponds the aerodynamic size of 645 nm, which might be closer to the 50 % cut-off size of the ACSM. However, as bigger particles are also transmitted through the lenses, even though not as efficiently as smaller ones, we chose to include the size bin #23. The higher size bin used in this study contributed 8.3 % to the total particle mass on a campaign-wide average. For that reason, we think the selected size range from DMPS is a reasonable approximation to the sizes measured by ACSM. A sentence was added to the manuscript to refer the associated uncertainty when selecting the bins for comparison with ACSM data.
The text was modified to:

P6. L161. Even though there is an associated uncertainty on the selected size range, since it depends e.g. on the calculated density and the ACSM transmission efficiency is not 100 % for all size ranges, the resulting PMC was considered as a reasonable approximation to the one estimated from ACSM and MAAP measurements.

L180-185. This is also related to the point above. The mass concentration calculations for ACSM seem circular. If the first density estimate used to convert SMPS volume to mass is based on an

equation which uses mass estimates of ACSM and if those mass estimates relative to BC are not correct, then the estimated density and SMPS mass concentration are not correct, so the ratio of (SMPS-BC) mass to ACSM mass is not correct. Please explain why you think this calculation is correct.

We understand that this calculation seems circular. However, the density value is governed by the relative contributions of the ACSM species (eBC monthly contribution was from 10-30 %, see Fig. 3). The relative contributions have a much smaller uncertainty than the absolute concentrations. The monthly median density of NR-PM$_1$ ranged from 1.27 to 1.48 g cm$^{-3}$ for both corrected and uncorrected density, with a median density of 1.34 and 1.35 g cm$^{-3}$, respectively. Also, if we consider that the maximum eBC monthly contribution to density was 30 %, the monthly median density when BC was maximum differed about 10 % from the median monthly density obtained in our calculations. As already mentioned, one large source of uncertainty in density calculations is the density of organics as they constituted more than half of the mass most of the time.
The text was modified as:

P6. L188. This approach provides a reasonable estimate of NR-PM$_1$ atmospheric concentrations. However, it has an associated uncertainty, which is expected to be minor considering that it is mostly determined by the particle density used in the correction and the particle components are both in nominator and denominator in the density Eq. (1).

L248. It's mentioned that the 4.5 yr dataset might not be long enough for trend analysis. How far back are similar data available? It seems some measurements are available since 2013. Can these two datasets be combined for just the trend analysis?

The Supersite station was established in 2015. The previous measurements performed in Helsinki before that refer to different environments.

L307. The seasonal explanation of VC doesn't match the monthly behavior as shown C2 ACPD Interactive comment Printer-friendly version Discussion paper in Fig. 5. It appears that VC is high during summer and low in winter. Why do authors say that ventilation was low in August? Add VC after 'ventilation coefficient' to define it since it's only discussed in SI.

According to our long-term data, VC is higher from May to July, which results in increased VC during summer (Fig. S7). However, the median VC decreases sharply in August, even though higher values are still observed according to the percentiles represented in Fig. 5. The most plausible explanation for this observation is that cold temperatures are already observed in August from late-evenings to early-mornings. In fact, the lowest VC were observed during summer from 11pm to ~10am (Fig. S7), which coincides with the morning rush-hour. For that reason, we think that the lowest VC during early morning can play a role on the measured eBC and partially explain the August maximums observed for this atmospheric constituent during the studied years. The VC was defined in Sect. 2.2.5 (L171).

P10. L311. The elevated concentration of eBC in August might be partly explained by poorer dispersion as the VC characterizing atmospheric dilution clearly decreased in this month (Fig. 5c). The VC is higher in June and July but decreases in August reaching the lowest median values of the year, which is probably explained by the cold temperatures already observed in August from late-evenings to early-mornings. However, the fairly high eBC concentrations in June and July are still surprising, especially considering the diminishing of traffic during holidays, and additional studies are needed to investigate this phenomena further.

L308. So what's the source of high BC during holiday season/summer? Long range transport?

The elevated eBC concentrations during holiday season/summer are puzzling. They can be a consequence of the lensing effect described in L317-319. However, they might as well be real but a prominent source of eBC has not been identified in Northern Europe. The poorer atmospheric dispersion during early mornings in summer might as well play a role. Additional laboratory studies involving a comparison between non-coated and coated eBC would be helpful to elucidate the influence from lensing effect on the measured eBC by MAAP, but these studies are currently missing. In terms of long-rang transport, we don't think that LRT was the source of eBC spikes during summer since inorganic secondary species, sulfate, nitrate and ammonium, that are largely related to LRT in Helsinki, were not elevated in summer. Also, the daily trends of eBC during the warmest months of the year revealed the expected peak in concentrations during traffic rush hours, which exclude many long-range transport sources such as forest fires (e.g. Fig. S12, note that $BC_{FF}$ and $BC_{WB}$ were determined from eBC measured by MAAP as described in Sect. 2.2.3).

P11. L324. However, additional studies are still needed to confirm this hypothesis or identify the existence of a prominent eBC source during summertime. According to our results, the source would likely be local since the background eBC concentration over the seasons remained relatively constant and the eBC daily trends during the warmest months showed the traffic rush hours peak, which excludes sources such as forest fires (e.g. Fig. S12).

L337. Cold start emissions during summer are still important although the duration of such conditions might be shorter.

We partly agree with the referee that cold start emissions can play a role also in summer. However, we think that the influence of mixing conditions is more important. The text was modified to:

P11. L345. The influence of mixing conditions is particularly evident during summer mornings when weak mixing conditions were observed (Fig. S7), causing high concentrations of $BC_{FF}$, even though the effect of cold start emissions cannot be totally excluded.

L348. Why can't the increased BCWB be from local sources?

The Supersite station is located in an area where residential biomass burning is expected to be negligible.

P3. L77. The residential area surrounding the street canyon uses mostly thermal energy from District heating, and therefore the local residential biomass burning contribution to PM is expected to be negligible.

L385. Too qualitative of a statement. Please indicate a number (either exact or say larger by ##%).

The text was modified to:

P13. L393. The seasonality of particle mass size distributions was as well investigated (Fig. 7b). Similar distributions were observed for the different seasons, with maximum mass concentrations occurring at around ~258 nm. These highest mass concentrations varied between 3.6 µg m$^{-3}$ in winter and 40.1 µg m$^{-3}$ in summer. The mass mode was also larger in summer and spring comparatively to the other seasons, particularly at sizes from 45 to 258 nm.

L400. Change exceed to exceeded

The text was modified as suggested.

L413. What was the concentration of BCff during this local pollution event? In the long-range transport case discussed in the following paragraph, it is indicated that during the local events relative contribution of BCff is not higher than during the long-range transport times and I find that surprising. Is that because its mass conc. is low relative to all the secondary species so the relative contribution stays more or less the same?

The median concentration of $BC_{FF}$ was 1.4 µg m$^{-3}$ during the local pollution episode and 0.8 µg m$^{-3}$ during the long-range transport episode. The relative contribution of $BC_{FF}$ was also higher during the local episode, corresponding to 13.7 % of $PM_1$, while during the long-range episode its contribution was 6.5 %. Since $BC_{FF}$ dominated eBC concentrations during the local pollution event (the $BC_{WB}$ relative contribution was 3.7 %), a source apportionment was not performed in Fig. 9 for clarity. It is surprising that the $BC_{FF}$ remains relatively constant when $PM_{2.5}$ increased in Fig. 10. The most probable explanation for that is the highest frequency of long-range pollution events comparatively to the local ones, which is then somehow hindering the increase in $BC_{FF}$ during local events. This is expected since the most frequent wind directions at the sampling site are from SE to SW (see L413-415), coinciding with the coordinates where long-range pollution events are particularly intense.

P14. L446. The relative contribution of $BC_{FF}$ remained relatively constant over the different $PM_{2.5}$ masses, varying between 7.8 % and 9.3 %. The almost constant $BC_{FF}$ contribution indicates that the local primary emissions from traffic exhaust are not usually the main source of high $PM_{2.5}$ episodes since primary exhaust emissions are characterized with the high fraction of $BC_{FF}$ (Fig. 9). However, the frequency of long-range pollution events is probably higher comparatively to the local ones due to the dominant SE to SW wind directions at the sampling site. These coordinates coincide with the ones where long-range pollution events are expected to be particularly intense, and their prevalence can then hinder the increase in $BC_{FF}$ during local events on a long-term analysis. The contribution of $BC_{WB}$ was the smallest for all $PM_{2.5}$ concentration bins.

L 446. I think it makes more sense to look at the PM2.5 fraction of PM10 rather than the ratio of PM10/PM2.5. Why was the ratio used in the analysis?

The Fig. was corrected as suggested and the text modified to:

P15. L458. Furthermore, the $PM_{2.5}/PM_{10}$ ratio decreased for $PM_{2.5}$ concentrations over 20 µg m$^{-3}$, with the decrease being particularly significant for $PM_{2.5}$ levels over 30 µg m$^{-3}$.

**Reviewer 2:**

Major Comments: In section 3.5, the author discussed the effects of local and long range transport by comparing a few factors for defining long range transport vs local formation. The author also used satellite data to support these arguments. I am curious whether the author did any examination based on certain tracer ions from previous studies to perform a more detailed the source apportionment of the data collected. For instance, have the authors analyzed m/z 82 signal to examine the fraction of the IEPOX-SOA? (Budisulistiorini et al., 2013, Hu et al., 2016), or m/z 60 to understand the concentration of levoglucosan or biomass burning (Bougiatioti et al., 2014)). With four and a half year data, the author should probably also use PMF to analyze the data and look for any information that the PMF may be able to provide. For instance, the author can look at the ratio between more aged organic components vs less oxidized, which may help distinguish aerosols from long range or local transportation.

A thorough source apportionment of organics is relevant to understand their sources and characteristics at the sampling site. Even though PMF is currently the most valuable tool for that purpose, it was not performed in this study since the focus was on the particle chemical composition in a long time-period. However, we characterized the organic sources based on their daily variations during different seasons. This was possible because organics increased during traffic-rush hours during winter-time but not during summer, which shows the prevalence of organics from traffic during winter and the dominance of SOA during the warmest period of the year. That evidence was confirmed by the DMPS measurements (Fig. 7). The m/z 60 could have been used as a tracer for biomass burning, but since Aethalometer can perform that source apportionment we relied on those results for the apportionment of BC from traffic and biomass burning. Furthermore, the residential area surrounding the street canyon uses mostly thermal energy from District heating and biomass burning is then expected to be only a minor source.

P3. L77. The residential area surrounding the street canyon uses mostly thermal energy from District heating, and therefore the local residential biomass burning contribution to PM is expected to be negligible.

It was a bit confusing when the author described the density conversion in line 150- 155. For instance, the author described a constant density of 1.5 g cm$^{-3}$ was used to convert mobility diameter to vacuum aerodynamic diameter, without specifying the reference. Then the author calculated the gravimetric density to be 1.42 g cm$^{-3}$. Why would the author not use 1.42 g cm$^{-3}$ to reconvert the mobility diameter to vacuum aerodynamic diameter again to make the results more accurate?

A clarification of the method used for comparison of ACSM with DMPS and density calculation was performed in the manuscript (see L92, L150 and the similar comment by Reviewer 1).

Minor Comment: L 135: Please define BC(FF) and BC(WB) when it first appear

The text was modified as suggested.